# Engineering Spatiotemporal Control in Vascularized Tissues

**DOI:** 10.3390/bioengineering9100555

**Published:** 2022-10-14

**Authors:** Astha Khanna, Beu P. Oropeza, Ngan F. Huang

**Affiliations:** 1Graver Technologies, Newark, NJ 07105, USA; 2Stanford Cardiovascular Institute, Stanford University, Stanford, CA 94305, USA; 3Department of Cardiothoracic Surgery, Stanford University, Stanford, CA 94305, USA; 4Center for Tissue Regeneration, Veterans Affairs Palo Alto Health Care System, Palo Alto, CA 94304, USA; 5Department of Chemical Engineering, Stanford University, Stanford, CA 94305, USA

**Keywords:** vascularization, tissue engineering, 3d bioprinting, biomaterials, cardiac engineering, extracellular matrix

## Abstract

A major challenge in engineering scalable three-dimensional tissues is the generation of a functional and developed microvascular network for adequate perfusion of oxygen and growth factors. Current biological approaches to creating vascularized tissues include the use of vascular cells, soluble factors, and instructive biomaterials. Angiogenesis and the subsequent generation of a functional vascular bed within engineered tissues has gained attention and is actively being studied through combinations of physical and chemical signals, specifically through the presentation of topographical growth factor signals. The spatiotemporal control of angiogenic signals can generate vascular networks in large and dense engineered tissues. This review highlights the developments and studies in the spatiotemporal control of these biological approaches through the coordinated orchestration of angiogenic factors, differentiation of vascular cells, and microfabrication of complex vascular networks. Fabrication strategies to achieve spatiotemporal control of vascularization involves the incorporation or encapsulation of growth factors, topographical engineering approaches, and 3D bioprinting techniques. In this article, we highlight the vascularization of engineered tissues, with a focus on vascularized cardiac patches that are clinically scalable for myocardial repair. Finally, we discuss the present challenges for successful clinical translation of engineered tissues and biomaterials.

## 1. Introduction

Cardiovascular disease (CVD) remains a major cause of morbidity and mortality all over the world, causing approximately 17.9 million deaths per year in the United States [1]. Heart failure, the inability of heart to provide sufficient blood flow to the body, is largely attributed to the non-regenerative cardiomyocytes that provide contractility to the heart [2]. Myocardial injury results in cardiac remodeling and adverse fibrosis, resulting in fibrotic scar tissue [3]. Patients with advanced failure resort to autologous or allogenic graft transplantation, often limited by factors such as donor site morbidity, non-availability of appropriate donor tissue in autologous transplantation, and the high probability of disease transmission associated with immunosuppression in allogenic transplantation [4]. However, failure of graft integration by inadequate vascularization post implantation can result in failure of graft. Tissue engineering to replace diseased tissues and organs is an alternative to autologous or allogenic graft transplantation. In the last two decades, tissue engineering has advanced the restoration and replacement of tissues and organs by using cells and biomolecules in three-dimensional (3D) biomaterials [4]. Although different avascular tissues such as bladder, cartilage, and epidermis have been successfully fabricated and progressed to clinical translation [5], tissue engineering of complex tissues that are thicker and vascularized have been limited in success due to the lack of a robust microvascular network [5]. Living cells typically reside within 200 μm of blood circulation for efficient oxygen and nutrient exchange for survival and functional bioactivity long-term. Due to the constraint of oxygen diffusion limit, most 3D engineered tissues of physiological architecture require adequate vascularization of tissue or perfusion source [5].

Strategies to induce vasculature formation in engineered tissues seek to mimic the re-vascularization process known as angiogenesis in response to tissue ischemia [6]. Understanding the mechanisms regulating angiogenesis can help discover clues to fabricate engineered tissues embedded with functional and mature vascular network [7]. The process of angiogenesis involves stimulation of quiescent vascular endothelial cells (ECs) from an existing vessel and their activation caused by high concentration of pro-angiogenic factors released by inflammatory cellular population as a signaling response to injury, hypoxia, or a combination of both. The ECs activate by proliferating and sensing the chemical gradients of soluble factors, resulting in stimulus directed elongation of new vessels via migration and secretion of soluble factors molecules that recruit perivascular support cells [7]. Perivascular cells, composed of capillary pericytes and smooth muscle cells in larger vessels, move towards the new vessels formed to cover the endothelium, imparting stability, inducing cell differentiation, and regulating permeability of vessel [8]. The critical processes include temporal regulation, spatial arrangement of the stimuli, crosstalk between cells and molecules, active remodeling, and organization of extracellular matrix (ECM). Dysregulation of any of the processes can result in abnormal development of new vasculature can occur due to disruption of the tightly regulated factors needed for angiogenesis.

Numerous engineering techniques have been developed to generate biomaterial constructs with unique spatial modifications and complex microarchitectures. In this review, we highlight the recent developments and state-of-the-art approaches in biofabrication techniques, including, but not limited to, electrospinning, micropatterning, and 3D bioprinting techniques for the generation of spatially defined biomaterials of optimal geometrical and topographical characteristics for modulating proliferation, migration, and differentiation of cells in contact with engineered scaffolds (Figure 1). We also review the temporal control advancements focused on the controlled release of growth factors and drugs to recapitulate the unique dynamic features of the ECM to direct biological processes, such as stem cell differentiation and functional tissue regeneration. The advancements and limitations of current biofabrication techniques for spatiotemporal regulation in multiscale materials and biomimetic microenvironments are also discussed. As an example, the spatiotemporal regulation strategies for the generation of vascularized engineered cardiac patches are discussed. Finally, we conclude by providing a perspective on the future challenges and opportunities in the development of biomaterials for tissue engineering applications.

## 2. Temporal Biology of Angiogenesis

During the process of angiogenesis, novel capillaries are formed from existing vasculature via the process of sprouting and intussusception. Several cellular functions occur during angiogenesis categorized into a phase of activation involving initiation and progression and a phase of resolution that includes processes of termination and maturation of vessel [9]. Biochemical processes such as ischemia or inflammation causes sprouting that promotes stimulatory autocrine and paracrine cytokines release, including the very potent vascular endothelial growth factor (VEGF) [10]. Leaky vasculature results from local basement membrane degradation and vessel endothelial cell’s migration, governed by a specialized tip cell [11]. Dll4-Notch lateral inhibition between neighboring endothelial cells in a feedback loop with VEGF–VEGF receptor (VEGFR) signaling is characterized to be the ‘central pattern generating’ (CPG) mechanism that results in the selection of migratory tip cells. The tip cells inhibit their neighboring cells, which are termed ‘stalk cells’ [11]. Several studies have demonstrated the part of ECM in the process of angiogenesis. The ECM creates physical scaffold necessary to maintain blood vessel organization and participates in biochemical and biophysical signaling transduction during angiogenesis. Specifically, collagen and fibronectin stimulate EC tubular morphogenic events [12]. Laminin facilitates endothelial cell tip formation and sprouting and are also critical for maintaining vascular homeostasis, and proteoglycans regulate endothelial cell migration to form new vessels [12]. During angiogenesis, matrix metalloproteinases (MMPs) degrade collagen and other ECM components, facilitating endothelial cell migration from pre-existing vessels towards angiogenic stimuli. Polymerization of plasma-derived proteins such as fibrinogen and fibronectin cause generation of a provisional ECM that facilitates endothelial cell extension in the environment. Stalk cells proliferation behind the tip cell, and mural cell populations recruitment induce the sprout’s elongation and formation of lumen [13]. The process of intussusception encompasses the existing vasculature remodeling through the protrusion and fusion of opposite vessel walls via split of an existing vessel to form a branched structure.

In comparison to sprouting, intussusceptive angiogenesis is rapid in capillary network expansion as it relies on reorganizing existing endothelial cells and not cellular division. Triggers such as hypoxia and injury facilitate angiogenesis resulting in immature microvascular networks through processes such as sprouting and intussusception, and pericyte invasion and subsequent cytokine-mediated cell induction that stabilizes the vessel walls and basement membrane followed by remodeling [13]. The premature vessel network is further remodeled to form an efficiently perfused vascular bed for oxygenation of the tissue. Vessel ablation of this nature is regulated by hemodynamic signals from local vasculature as endothelial cells integrate with nearby perfused network or undergo apoptosis causing vascular regression without pro-survival stimuli including shear flow and growth factor gradients [14].

Tissue engineering aims at inducing an angiogenic response from host ECs to utilize the natural capacity of our system for tissue vascularization through angiogenesis and regenerative techniques employed in specific applications, such as an infarcted heart. At the cellular level, mechanical and chemical signals govern the internal signaling effect and consequent biological responses including migration, proliferation, and differentiation. Hence, temporal conjugation of biomolecules in a tissue engineered scaffold plays a major role in development of mature vascular network. Commonly, non-covalent adsorption of growth factors and other molecules incorporated into engineered scaffolds is employed. Thus, growth factor release depends on the affinity of the molecules with the scaffold material or regulated through molecular diffusion kinetics. This approach is advantageous when scaffolds are employed for the controlled release of molecules to organs or tissues.

## 3. Growth Factors Regulation in Angiogenesis

Angiogenesis is controlled by cell–cell and cell-ECM interactions through crosstalk between VEGF and Notch signaling mechanisms. Novel vascular structures are regulated by the surrounding cells and modulated by secretion of platelet-derived growth factor (PDGF) and VEGF secreted by ECs and vascular smooth muscle cells [15]. Among the factors that impact EC activation status are proteins called cytokines. A tissue can dictate the cellular response to a given cytokine. Hence, cytokines are considered as specialized symbols in intercellular interaction. This interaction is influenced by three factors, including the concentration of other cytokines in the environment; chemical and biological interactions between ECM, cells, and cytokines; and the cytoskeleton [16].

The signal protein most commonly studied to influence angiogenesis are VEGF, acidic fibroblast growth factor (aFGF), and basic fibroblast growth factor (bFGF) [17]. VEGF and FGF-2 have been studied in vitro to positively regulate several endothelial cell functions, that includes cellular proliferation, migration, extracellular proteolytic activity, and tube formation [17]. In addition, although a myriad of factors have been demonstrated to be active in the experimental setting, they are not all relevant to the endogenous regulation of new blood vessel formation. On the list of molecules that are active during the phase of activation, VEGF meets most of the criteria of a vasculogenic or angiogenic factor.

Angiogenesis regulators may act either directly on ECs or indirectly by inducing the production of direct-acting regulators by inflammatory and other non-EC populations. Thus, in contrast to VEGF and FGF-2, which are direct endothelial cell mitogens, the cytokines transforming growth factor-β (TGF-β) and tumor necrosis factor-α (TNF-α) have been studied to inhibit EC growth in vitro and are therefore direct-acting negative regulators [18]. However, both TGF-β and TNF-α are angiogenic in vivo, and it has been demonstrated to induce angiogenesis indirectly by stimulating the production of direct-acting positive regulators from stromal and chemoattracted inflammatory cells; hence, TGF-β and TNF-α are considered to be indirect positive regulators [19]. TGF-β has also been proposed to be a potential mediator of the phase of resolution due to its capacity to inhibit endothelial cell proliferation and migration directly, reduce extracellular proteolysis, and promote matrix deposition in vitro. In vitro, TGF-β has also been studied to promote the organization of single endothelial cells embedded in three-dimensional collagen gels into tubelike structures, further signifying its role in the phase of resolution [18].

Other cytokines that have been studied to regulate angiogenesis in vivo include HGF, EGF/TGF-α, PDGF-BB, interleukins (IL-1, IL-6, and IL-12), interferons, GM-CSF, PlGF, proliferin, and proliferin-related protein. Angiogenesis can also be regulated by a variety of noncytokine or nonchemokine factors, including enzymes (angiogenin and PD-ECGF/TP), inhibitors of matrix-degrading proteolytic enzymes (TIMPs) and of PAs (PAIs), extracellular matrix components/coagulation factors or fragments (thrombospondin, angiostatin, hyaluronan, and its oligosaccharides), soluble cytokine receptors, prostaglandins, adipocyte lipids, and copper ions. The roles of these bioactive molecules are summarized in Table 1.

### Combinatorial Regulation Chemical Factors in Engineering Vascularized Tissues

Biomaterial matrices functionalized with angiogenic growth factors have been extensively researched to promote vascularization. A myriad of angiogenic growth factors such as VEGF, PDGF-BB, bFGF, hepatocyte growth factor (HGF), insulin-like growth factor (IGF), and TGF-β have been widely studied to promote vascularization in pathological disease models [26]. All key angiogenic growth factors (VEGF, FGF-2, IGF, HGF, PDGF-BB, and TGF-β1) bind to specific sites in the ECM; their release kinetics are based on their binding affinity and the proteases action to cleave the ECM or the ECM-binding growth factor domain [26]. It has been demonstrated through in vitro and in vivo studies that insufficient angiogenic growth factor exposure can inhibit angiogenesis, and subsequently, overexpression of growth factor can inhibit the function of vascular smooth muscle cells and pericytes population and form immature and unstable vessels [27]. The dose and duration of growth factor release has been studied to play a critical role in therapeutic applications. Several strategies have been employed to control the release of growth factors from biodegradable scaffolds. For example, angiogenesis has been enhanced using heparin or heparan sulfate-mimetic molecules covalently crosslinked with the collagen type I scaffold via 1-ethyl-3-dimethyl aminopropyl carbodiimide (EDC) and N-hydroxysuccinimide (NHS) for release of heparin-binding growth factors [28]. Further, angiogenesis has been studied to be enhanced by the combination of VEGF and FGF with a heparin-immobilized scaffold compared with a single growth factor molecule. Biomaterials have also been functionalized using surface modification strategies or heparin-binding ECM domain addition. For example, sequestration of multiple growth factors (VEGF-A165, PDGF-BB, and BMP-2) can be achieved using a fibrin matrix covalently crosslinked with multifunctional recombinant fibronectin (FN) fragments, including both its 12th and 14th type III repeats (FN III12-14) and FN III9-10 for enhanced angiogenic effects [29]. Angiogenic growth factors can also be altered for enhanced binding affinity to biomaterials for enhanced affinity with growth factors. Sacchi et al. achieved covalently crosslinking of fibrin hydrogels with VEGF fused to a sequence derived from α_2_-plasmin inhibitor (α_2_-PI_1–8_) for controlled VEGF release by enzymatic cleavage, that resulted in stable and functional angiogenesis [30].

Incorporation of short bioactive peptides onto 3D scaffolds has gained interest as an effective method to achieve vascularization. Several approaches have been employed to study the effects of the immobilized bioactive peptides on vascular network formation. Increased EC attachment, growth, and migration were achieved by incorporation of integrin to ECM derived short peptide adhesive sequences such as collagen (Arg-Gly-Asp (RGD)), laminin (e.g., Tyr-Ile-Gly-Ser-Arg (YIGSR) and Ser-Ile-Lys-Val-Ala-Val (SIKVAV)), and FN (e.g., RGD and Arg-Glu-Asp-Val (REDV)) that increased angiogenesis [31]. Hydrogel activation by functional RGD and REDV sequences in an elastin-like recombinamer-based hydrogel caused improved EC adhesion and in vivo angiogenic potential via general cell adhesion and specific endothelial cell adhesion.

Several strategies have been taken to deliver bioactive molecules from tissue engineered scaffolds that mimic those associated with angiogenesis. Although the delivery of single-factor soluble factors such as bFGF can induce EC proliferation [32], the delivery of combinatorial growth factors better mimic the complexity of the angiogenic process. Various studies have demonstrated controlled dose and duration of growth factor release from biodegradable materials. Heparin-binding growth factors, VEGF and FGF-2 delivered from heparin-immobilized scaffolds exhibited an increased degree of angiogenesis in comparison to individual growth factor response [33]. Multiple growth factors (VEGF-A165, PDGF-BB, and BMP-2) were sequestered using fibrin matrix covalently crosslinked with multifunctional recombinant fibronectin (FN) fragments (12th and 14th type III repeats (FN III12-14) and FN III9-10) and exhibited enhanced angiogenic effects in a mouse model of chronic wound healing [33]. In another example, Kuttappan et al. functionalized a nanocomposite fibrous scaffold with combinations of VEGF, FGF-2, and BMP2 for differential growth factor release [34] that resulted in increased tissue vascularization. Furthermore, since growth factor delivery based on scaffold degradation can lead to an initial burst release, it was shown that increasing the crosslinking density of gelatin could improve growth factor retention. Turner et al. achieved controlled release of VEGF or BMP2 based on the progressive proteolytic degradation of the scaffold using crosslinked gelatin microspheres. However, the non-specific degradation of the scaffold and non-uniform growth factor release necessitates the need for development of more advanced systems for optimal release [35].

Incorporation of different bioactive molecules in sequential layers of polymers; a technique called layer-by-layer (LBL), can be employed for sequential delivery of growth factors. The incorporation of bioactive molecules and action of matrix-degrading enzymes causes sequential delivery of growth factors. A Polycaprolactone (PCL) scaffold was developed with sequential layers of heparin and VEGF was developed. Long-term anti-thrombogenic effect of tissue engineered graft was achieved by initial burst release of VEGF, facilitated by ECM degrading enzyme metallopeptidase-2 (MMP-2) and controlled release of heparin [36].

An enzyme-sensitive linker to link pro-angiogenic molecules covalently to the scaffold has been studied to promote angiogenesis. Linking the linker sequence to a specific enzyme (e.g., MMPs, serine, or cysteine proteinases) regulates time-bound release as enzymes are produced by cells at specific times during differentiation or angiogenesis. Light, an external stimulus for smart drug-delivery platforms, has been studied in various biomedical applications including image-guided surgery, and the photopolymerization and -degradation of tissue engineering scaffolds, for the advantages of its noninvasive properties, high spatial resolution, temporal control, and simple to use [37]. Light-sensitive linkers have been used to covalently bind molecules, with UV or near infrared (NIR) light used to release “incorporated” biomolecules. Light-responsive delivery systems must possess high spatial and temporal regulation over drug release, employ nonionizing radiation, formed of biocompatible materials, and flexible to be tailored to the needed application [38].

Encapsulation is another technique for controlled release of bioactive molecules. It has the advantage of providing protection for growth factors, increasing their half-life. Studies have been performed to design a scaffold patterned with composite microspheres, with the spatiotemporal release of proteins [39]. Lai et al. employed the technique of encapsulation via nanofibers and gelatin nanoparticles to form a scaffold for sequential release of VEGF, PDGF, FGF, and EGF (epithelial growth factor). This resulted in enhanced endothelial cell proliferation and development of vascular networks [40]. Various approaches have been employed for controlled local delivery of angiogenic growth factors, however, the limitation of their inherent inability to control the geometric architecture of vascular networks needs to be addressed for optimal 3D tissue construction. These techniques can be employed to control growth factor delivery recapitulating the temporal pattern observed in physiological angiogenesis, but with limited complexity. Despite the promise growth factor delivery or bioactive-peptide-guided vascular network formation, these approaches still lack the control network geometry, for generation of a spatially controllable 3D mature vascular network. Therefore, advancements in fabrication technologies below aim to fabricate spatially controllable 3D vascular networks using scaffolds.

## 4. Spatial Control in Engineering Vascularized Tissues

Besides temporal regulation of growth factors, angiogenesis is strictly regulated by spatial signals to govern vessel sprouting and maturation. Physiological cues including inflammation and ischemia induce release of growth factor molecules, cytokines to create a gradient within the extracellular matrix domain that results in generation of a spatially controlled rearrangement of neovessels. Tissue engineering strategies seek to develop systems with spatial control, such as direct cellular patterning using 3D bioprinting, electrospinning and soft lithography for more precisely controllable neo-vessel formation. These systems have been highlighted below.

### 4.1. Three-Dimensiona; Bioprinting

Three-dimensional bioprinting is a multidisciplinary approach to spatially pattern cellular and biological components by employing a layer-by-layer process to deposit and generate 3D organ analogs and tissues platforms. Researchers have widely employed 3D bioprinting to generate 3D constructs with vascular networks for creation of more geometrically complex tissue structures. Bioprinting utilizes two manufacturing approaches of direct and indirect printing to create tissue constructs. Direct printing includes printing of bio-ink droplets (cell-laden hydrogels) containing cellular and extracellular components into designed vascular network structures. In contrast, indirect printing involves a cell-free scaffold or component bioprinted with cell-laden hydrogel layers. Using these methods, combination of cells, biomaterials, and growth factors can produce complex constructs with spatial organization with dimensional micron-sized channels and pore sizes to direct angiogenesis.

The most commonly employed bioprinting techniques are based on inkjet, extrusion, and lasers methods (Table 2). Inkjet bioprinting involves layer-by-layer dispersion of bio-ink droplets on a construct using a thermal or piezoelectric actuator. In inkjet bioprinting, the printhead is placed over the printing bed, followed by generation of a 3D tissue using bioink droplets created by thermal, electrostatic, or piezoelectric inkjet bioprinters [41]. This approach has the advantage of generating picoliter-scale drops with a ~30 to 60 μm printing resolution. It utilizes crosslinking molecules with hydrogels having rapid gelation characteristics for generation of organized networks. For example, alginate-based bio-inks can be printed into a calcium chloride solution as they rapidly crosslink. This technique has been used to create 200-μm diameter vessels. Cui et al. demonstrated effective simultaneous printing of ECs and fibrin-based vascular networks. The aligned ECs proliferated to form a confluent tubular form within the printed channels within the fibrin scaffold after 28 days of culture [42]. This approach has the advantage of low cost due to the potential to adapt regular printers and for printing multiple cell types. The thickness of constructs printed using inkjet methods are limited by weak structural support due to low concentration of hydrogel.

Extrusion-based bioprinting encompasses layer-by-layer printing of bio-ink by employing a syringe and piston for dispensing through nozzles on a microscale level. Extrusion-based approaches employ relatively higher amounts of hydrogels such as alginate and Pluronic F-127 for the generation of stable 3D cellular constructs. Millik et al. optimized an advanced extrusion system and bio-ink blend for generation of highly organized and perfusable cell-loaded microvasculature [43]. Tubes produced with a wide range of diameters (500–1500 μm) and wall thicknesses (60–280 μm) using the co-axial system. Gao et al. recently constructed a coaxial extrusion system for concurrent flow of calcium solution (interior) and alginate solution (exterior) [44]. Hollow and high strength calcium alginate filaments of cell-laden, 3D hydrogel structures were successfully fabricated with microchannels that were perfusable. One of the limitations is the sub-optimal mechanical stability and structural integrity, for printing clinically scalable tissue constructs. In a study conducted by Kim et al., an advanced 3D printer known as an integrated tissue-organ printer (ITOP), was employed to generate stable and multiform human-scale tissue constructs. [45]. The ITOP patterns multi- cell-laden composite hydrogels composed of gelatin, fibrinogen, hyaluronic acid, and glycerol and present a PCL polymer (of high strength) and a sacrificial Pluronic F-127 hydrogel with strong mechanical characteristics. The application of ITOP to generate a human-scale mandible, calvarial bone, cartilage, and skeletal muscle was successfully performed in vivo with mature and perfusable tissue formation [46]. The ITOP has been demonstrated to allow advanced 3D bioprinting and generation of clinically translatable tissues.

Lastly, laser-assisted bioprinting is an effective method for printing precise microvasculature, although fewer studies have been discussed on this. Laser-based bioprinting can be conducted using photopolymerization or laser-induced forward transfer method. Although this technique is costly, it can print cells at very high resolution obviating the exposure to high shear stress Wu and Ringeisen [47] employed laser bioprinting for generation of branch/stem structures with HUVECs followed by culture of human umbilical vein smooth muscle cells on the printed HUVEC constructs. The formed microvasculature possessed two stems and a stable lumina recapitulating the microvascular network.

**Table 2 bioengineering-09-00555-t002:** Comparison of bioprinting methods for fabrication of vascularized engineered tissues.

Bioprinting Technique	Bioprinted Cellular Types	Vascularization Application	Limitations	Ref
Inkjet Based Bioprinting	Human umbilical vein endothelial cells (HUVECs) Rat Smooth muscle cells (SMCs)	Heterocellular tissue patterned with <100 µm droplets. Employs thermal, electromagnetic, or piezoelectric strategy for deposition of “ink” droplets Rapid printing speeds with high resolution. Potential to print biomaterials with low viscosity. Availability and ease of using multiple bioinks. High-cellular viability and relatively less expensive	Low material viscosity (<10 Pa.s). Lack of precision with respect to droplet size. Requirement for low viscosity bioink. Nozzle clogging and cellular distortion due to high-cell density. Low mechanical strength. Inability to provide continuous stream of material	[48]
Extrusion Based Bioprinting	Human umbilical vein endothelial cells (HUVECs) Human umbilical vein smooth muscle cells (HUVSMCs), human bone marrow derived mesenchymal stem cells (hMSCs) Mouse embryonic fibroblasts (MEF)	Coaxial extrusion enables >80 cm long vascular conduits of lumen diameter 1520 µm to be developed. Heterogenous tissues constructs can be created (>1 cm in thickness and 10 cm^3^ volume). Multicellular spheroids (>400µm diameter) are bioprinted and double layered small diameter conduits of diameter 2.5 mm. Potential for printing biomaterials with high cellular densities (higher than 1 × 10^6^ cells/mL). Continuous stream of material can be generated High viscosity bioinks such as polymers, clay-based substrates can be printed.	Low printing resolution (>100 µm) and slow printing speeds. Loss of cellular viability and distortion of cellular structure due to the pressure to expel the bioink.	[49]
		Microchannels of width > 100µm can be obtained. Fast printing speeds and potential to print biomaterials with broader viscosity gradient (1–300 mPa/s). High precision and resolution (1 cell/droplet) can be achieved. High density of cells can be printed- 108/mL	Takes longer to generate—need to prepare reservoirs/ribbons. Low cellular viability compared to other techniques. Thermal damage can cause loss of cells. Intense UV radiation needed for SLA for crosslinking process. Large amount of materials needed and high cost. Longer post processing time and fewer materials have been found SLA-compatible.	[50,51]

Abbreviations: HUVEC (Human Umbilical Vein Endothelial Cells); hMSCs (Human Mesenchymal Stem Cells); MEF (Mouse Embryonic fibroblasts); NIH ST3 (Murine Fibroblasts); SMCs (Smooth Muscle cells); SLA (Stereolithography) bioprinting.

#### Multi-Material Bioprinting

For printing heterogenous and complex tissues, traditional bioprinters have the limitation of deposition of single bio-ink formulation from the single nozzle or postprocessing through layer-by-layer deposition. Multi-material bioprinting integrating multi-material platforms for bioprinting heterogenous, multicellular and functional tissue constructs has recently gained attention. It is advantageous for concurrent or sequentially depositing different materials such as cell-laden hydrogels or extracellular matrix structures, sacrificial materials, and polymers for scaffolding with hierarchical microstructure. These have been very promising for native tissue biomimicry. A myriad of multiple-head multi-material bioprinting strategies have been established for the constructing multicellular and zonally stratified organization of blood vessels for cell depositing on exogenous or other biomaterials. Tan et al. employed multi-head multi materials bioprinting for generation of generated concentric and self-supporting tubular structures. The group used two extrusion-based bioprinting printheads; first printhead was employed to extrude the alginate-xanthan gum hydrogel blend bio-ink in a circular pattern and, second printhead was designed for extruding crosslinker solution into inner-side of the printed circular pattern for high mechanical stability of tube wall [52]. Another group, Campbell et al., extruded several hydrogels using integration of a single printhead equipped with a selector valve to switch between separate syringe pumps, to allow sequential and controlled biofabrication of heterogeneous, multilayered and multicellular complex vascular tissue structures [53]. Pre-crosslinked cell-laden alginate-collagen blends of specific viscosity used as bio-ink with EC-laden bio-ink. This was successfully deposited and sequentially surrounded by extrusion of SMC-laden bioink.

Besides using bio-ink-based bioprinting, a scaffold-free approach using multicellular spheroids and cylinders have been employed using for generating vascular constructs [54]. Forgacs et al. demonstrated the application of scaffold-free multi-material bioprinting multilayered and multicellular vascular tubes, as one printhead was employed for deposition of agarose rods (molding template) and other was a pre-set extruded multicellular spheroid or cylinder [55] The fabrication of vascular tubes with linear and bifurcated geometries has been employed. Furthermore, a double-layered vascular construct composed of inner layer (HUVSMCs cylinders) and outer layer (human dermal fibroblast (HDF) cylinders) was created to recapitulate layers of native blood vessels --tunica media and tunica adventitia, respectively. Kucukgul et al. generated a biomimetic macrovascular construct using an algorithmic model and successfully performed scaffold-free bioprinting of aortic tissue constructs on capillary-based extrusion. Human aorta of mouse embryonic fibroblast (MEF) aggregates and agarose structures from two separate printheads were imaged [56].

Microfluidic multi-material bioprinting techniques have also been employed for the fabrication of vascular structures [57,58]. Attalla et al. deposited several viscous hydrogels from a multiaxial microfluidic printhead to engineer tubular constructs with cell-laden bioinks and the crosslinker solution added using needles in the microfluidic chip and concentrically dispensed from the nozzle [57]. Zhou et al. bioprinted a vessel-like tubular construct by employing a capillary-based microfluidic printhead where cell-laden alginate was released from six outer channels of the multi-barrel capillary nozzle, and CaCl_2_ was released from the central channel for crosslinking the bioink solution into a lumen [58]. Feng et al. developed a multicomponent bioprinting platform for biofabrication of artificial vessels where two alginate-based bioinks encapsulated with HUVECs and embryonic rat cardiomyocytes were extruded from the coaxial microfluidic printhead on a rotating material resulting in layer-by-layer fabrication of concentric ring structure [59]. Table 3 provides a comparison of various multi-material bioprinting approaches for the fabrication of vascular tissues.

Despite its capability for extruding meter-long vascular-like constructs, this approach is limited in the recapitulation of branched vascular tissues. Microfluidic multi-material bioprinting approaches, particularly the ones combined with coaxial nozzles, have the capability to recapitulate the native vascular tissues. Therefore, microfluidic printheads should be studied further to support the creation of freeform, multiscale vascular constructs in integration with the embedded bioprinting technique. Multi-material bioprinting platforms emerge as a powerful tool for replication of heterocellular and hierarchical composition of living tissues and organs needed for successful translation of engineered tissues and organs for clinical applications.

### 4.2. Electrospinning

Electrospinning employs electrical forces to generate nanofibers of a wide range of materials [74]. Nanofibers are generated using a polymeric solution injected from the syringe to a center of high electric field. As electrostatic forces get higher than the surface tension of the polymeric solution, this results in the formation of a Taylor cone with rapid acceleration of narrow jet towards the target (collector), connected to the ground with opposite charge [74]. In the past decade, electrospinning has been explored for generation of nano-fiber based microvasculature. This approach allows for fine control over diameter, porosity, and degradation rate. In addition, this technique results in formation of fibers having diameter similar to native ECM (50–500 nm) and mimic natural topographical cues [75]. Bioink deposited remade nanofibers have been designed for macroscale hydrogel constructs with nanoscale spatial control. Integration of electrospinning with 3D fiber deposition has been used to generate multiscale scaffold of a PEG/poly (butylene terephthalate) (PBT) block copolymer of native microarchitecture. In vitro studies revealed long term viability and high metabolic rate of human mesenchymal stromal cells cultured on these scaffolds arranged aligned along the scaffold [76]. This study demonstrated the application of multiscale, multi-compound, and multifunctional engineered tissues to recapitulate the complex native tissues.

In another study, Kim et al. studied endothelial differentiation of induced pluripotent stem cells (iPSCs) with topographically aligned 3D electrospun PCL scaffolds to produce iPSC-derived ECs (iPSC-ECs) [77] (Figure 2). The group reported enhanced gene expression of EC phenotypic markers CD31, CD144, and nitric oxide synthase within 3D scaffolds, compared to on 2D PCL films. Parallel-aligned vascular-like networks cultured with iPSC-ECs displayed 70% longer branch length in comparison to randomly oriented scaffolds (Figure 2C) This study revealed ability of fiber topography mechanism for modulating vascular network-like formation and patterning of structures. Wanjare et.al engineered cardiovascular tissues and demonstrated a dominant role of scaffold anisotropy to maintain human induced pluripotent stem cell-derived organization of cardiomyocytes (iCMs) and contractile function [78]. In another example, Kenar et al. blended poly(L-lactide-co-ε-caprolactone) (PCL) with collagen and hyaluronic acid and designed a micro-fibrous composite scaffold with enhanced length of vasculature [79]. Furthermore, Cui et al. demonstrated improved pre-vascularization of constructs by pre-seeding HUVEC on LBL aligned (PCL)/cellulose nanofiber matrices pre- implantation [80]. In vivo, the aligned fiber matrices integrated with the host vasculature. Electrospinning has the potential to produce matrices capable of mimicking ECM for enhanced in vivo angiogenesis through the fiber spatial organization, further highlighting the scope of the technique to improve vascularization.

### 4.3. Patterning of Bioactive Molecules

Biofabrication methods can be employed to pattern bioactive molecules within scaffolds to emulate biochemical gradients present in natural angiogenesis and promote in situ vascularization via integration with the host vascular network. Owing to its potent effect on angiogenesis, VEGF is the most commonly employed growth factor for the patterning of scaffolds. For example, Alsop et al. printed VEGF in a spatially defined manner onto a collagen-glycosaminoglycan scaffold by employing photolithography. The group reported high cell infiltration and immature vascular networks [81]. Directional vessel growth via pre-defined release of VEGF have been induced by using hydrogels [82]. Promotion of aligned vasculature as the hydrogel is printed parallel to existing vasculature and not perpendicular depicts the precise control. Studies have also incorporated growth factor combinations into the scaffold material for recapitulating the sequential stages of angiogenesis. Combinations of VEGF, FGF, and BMP2, and with VEGF and Angiopoietin have been studied to improve angiogenesis [83]. Even with the use of multiple growth factors, the platforms are still relatively basic compared to the complex process of angiogenesis. Incorporation of bioactive molecules in the scaffold material does not adequately ensure its spatial localization due to diffusion and burst release. Therefore, for development of functionalized decellularized scaffolds, several strategies have been employed to use heparin via endpoint attachment. This approach has been successful for binding and controlled release of heparin-binding growth factors, for example, VEGF, for enhanced angiogenesis [84]. Another group printed biodegradable polymer scaffold with concurrent zones of VEGF and VEGF inhibitors and achieved spatially restricted signaling [85].

Patterning bioactive molecules within scaffolds is another way for mimicking biochemical gradients or promoting in situ vascularization as a result off integration with the host vascular network. This spatial micropatterning approach to formation of vascular network can achieve a spatial resolution of less than 10 μm. This technique can engineer spatially organized ECs using microfabrication technologies, such as soft lithography and photopolymerization [86]. The steps to soft lithography include (1) design of pattern; (2) photomask and master fabrication; (3) fabrication of polydimethylsiloxane (PDMS) stamp; and (4) micro- and nano-structure fabrication employing the stamp. Raghavan et al. employed soft lithography techniques on microfabricated PDMS templates with intended geometries [87]. Spatially arranged endothelial cords were formed using a suspension of ECs in collagen gel introduced into the channel and stimulated with VEGF and bFGF. Baranski et al. studied micropatterned EC cords with human hepatocytes implanted into nude mice. Implanted cords directed rapid vascularization, and anastomosis of the cords with the host vasculature. [88] Another strategy for the controlled release of bioactive molecules is encapsulation. Encapsulation also provides a layer of protection for growth factors, improving their short half-life. Minardi et al. achieved the spatiotemporal release of proteins by patterning the scaffold with composite microspheres [89]. In addition, Nazarnezhad et al. produced a scaffold to achieve concurrent release of VEGF, PDGF, FGF, and EGF by encapsulation via nanofibers and gelatin nanoparticles [90]. Due to gradual release of these growth factors sustained for 45 days, enhanced endothelial cell proliferation and development of vascular-like structures was observed.

Along with growth factors, scaffolds have been functionalized with peptides to induce vascular growth and network formation. Covalent binding of peptides to the scaffold helps to pattern peptides on to surfaces and scaffolds with relative resilience to processing. These peptides possess greater flexibility than growth factors and can incorporate angiogenic domains. Lei et al. micropatterned SVVYGLR peptide strips on polymer surfaces using photolithography. Directional regulation and morphogenesis of ECs cultured onto 10, and 50 μm aligned peptide strips generated tubular structures [91]. Chow et al. employed peptide-PCL conjugates with selective affinity for glycosaminoglycans (GAGs), in combination with sequential electrospinning techniques, to direct the spatial patterning of GAGs all through the scaffold [92]. This helps to protect the bioactivity of the GAGs and native ECM, for a more clinically translatable tissue structure. These studies underline the importance of spatial organization in vascularization strategies. Angiogenesis has been studied to be modulated by micropatterning strong mechanical forces, as convex part of micropatterned vessel walls allows for preferential blood vessel formation. Huang’s group demonstrated parallel-aligned micropatterned channels as well as nanopatterned collagen scaffolds to promote the organization and migration of ECs [93]. Laminar flow applied to EC-seeded parallel-aligned nanofibrillar collagen scaffolds and orthogonal to the direction of collagen patterning, the cellular population preferentially remained organized along the spatial patterning direction.

Patterning of biomolecules shows great promise for promoting vascularization in engineered scaffolds and tissues owing to the high spatial precision of micropatterning techniques, Biomolecules such as peptides can be incorporated into micropatterned or nanopatterned scaffolds (Table 4). The limitations of random distribution of growth factors have been addressed by biomolecules encapsulation pre-patterning of scaffolds. The use of nanoparticles is being explored for strict control on biomolecule patterning and release. In addition, as the size-scale of spatial micropatterned substrates is limited, hence, strategies that allow generation of larger-scale vascular networks are actively studied.

## 5. Spatiotemporal Regulation of Engineering Vascularized Cardiac Patches

### 5.1. Vascularized Cardiac Patch with Temporal Regulation

#### 5.1.1. Engineering Vascularized Patch with Temporal Regulation In Vitro

Tissue engineering encompasses principles of engineering and biology for the generation of living tissues studied for drug screening, disease modeling, and therapeutic regeneration. Techniques to reprogram human somatic cells into iPSCs and differentiation into cardiomyocytes and other cardiac cells have been extensively studied to be efficient, which has led to accelerated progress towards the generation of engineered human cardiac muscle patch (hCMP) and heart tissue constructs [101,102]. Traditional methods for hCMP fabrication involve suspending cells within biocompatible material scaffolds or culture of two-dimensional sheets to form multilayered constructs. Recently, spatiotemporal techniques such as micropatterning and three-dimensional bioprinting have been employed to generate hCMP architectures at unprecedented spatiotemporal resolution [102]. One limitation of hCMP-based strategies for in vivo tissue repair is inadequate scalability, poor integration and engraftment rate, and the lack of functional vascular networks. Therefore, cardiac patches must be designed to allow assimilation with the host myocardium and synchronization. Porous scaffolds have been widely studied as a promising biomaterial as the architecture provides appropriate directions for cells for matrix penetration. This strategy can induce adequate rate of biomaterial degradation for new tissue reconstruction with improved nutrient supply and electrophysiological mediated integration. For instance, improved in vivo vascularization within the patch was achieved using VEGF-containing scaffolds [103]. Pre-vascularization of cardiac patch scaffolds has also been studied to improve mass transport. MSCs, through the release of angiogenic factors have been shown to support the formation of microvessels and their structure. ECM nanofibers and MSCs were shown to promote vascular constructs as a sheet when co-cultured with ECs. Shevach et al. suggested decorating decellularized matrices with gold nanoparticles and nanowires for improved electrical coupling, presenting stronger contractile force and lower excitation frequency [104].

Induced pluripotent stem cells (iPSCs) have gained attention as a key component of cardiac tissue engineering for understanding of cardiovascular disease mechanisms, drug responses, and developmental processes in human 3D tissue models. A wide range of iPSC-derived cardiac spheroids, organoids, and heart-on-a-chip models have been developed since the very first engineered tissue was fabricated more than two decades ago. The iPSC-derived cardiovascular cells can be differentiated by soluble factors (e.g., small molecules), extracellular matrix scaffolds, and exogenous biophysical maturation cues [105]. Efficient cardiomyocyte (CM) differentiation protocols, in combination with advancements in engineered biomaterials and organ-on-a-chip technology, have led to a variety of in vitro cardiac tissue models, ranging from spheroids and organoids to transplantable cardiac patches and 3D-bioprinted hearts. Protocols for differentiation into other major cardiovascular cell types (iPSC-derived endothelial cells, iPSC-ECs and iPSC-derived vascular smooth muscle cells) have been extensively studied in the past decade [106,107,108,109]. The major advantage of incorporating various iPSC derived cardiovascular cell types is to generate a more physiological construct, as demonstrated by the improved structural and functional maturity of multi-cell type microtissues. Multi-cellularity also enables researchers to study pathogenic mechanisms and drug responses to a specific cell that provides a versatile tool to study intercellular communication mechanisms (e.g., paracrine or contact-mediated). Despite the promise of CMs derived from human induced pluripotent stem cells (hiPSCs), these cells are found to be functionally immature and exhibit fetal-like features. To improve CM maturation, Mummery et al. showed that the tri-cellular combination of hiPSC-derived CMs, cardiac fibroblasts, and iPSC-ECs could enhance CM maturation in scaffold-free, three-dimensional microtissues [105,106]. Integration of engineered biomaterials with various microfabricated devices, stretch, and electrical circuits with conventional 2D approaches is being studied to overcome these limitations [110,111].

Biological materials have been extensively used as drug delivery vehicles. Various polymeric materials have been successfully studied to encapsulate or entrap biomolecular components resulting in low-dimension particles (microns to sub-nano scale). Such particles enable the delivery of soluble and insoluble bioactive molecules to the target site, providing enhanced stability, drug half-life, pharmacokinetics, and drug specificity. Drug delivery platforms could also be fabricated from biomaterials incorporated with delivery agents. Different fabrication methods and chemical formulations could be designed for tailoring the mechanical properties of biomaterials. These matrices are used for the controlled release of drugs, depending on factors such as degradation/erosion rate, triggers, or environment factors. Neighboring cells in the natural microenvironment communicate with each other via paracrine pathways and factors mediated by proteins, small RNA molecules, and extracellular vesicles (EVs). ECM acts as a reservoir of signaling components, the incorporation of proteins and protein-binding features into biomaterials could mimic ECM function and induce cellular responses, such as cell proliferation, migration, and differentiation [112]. Therefore, ECM-based biomaterials are potential candidates as advanced drug delivery systems for spatial–temporal presentation and delivery of therapeutic drugs, imperative for endogenous cardiac tissue regeneration and to restore functionality post cardiac injuries.

#### 5.1.2. Engineering Vascularized Patch with Temporal Regulation In Vivo

For localized and temporary delivery of bioactive molecules, controlled and sustained release systems have been designed. Polymeric materials, forming 2D and 3D matrices, have been extensively studied. As endogenous cardiac regeneration strategies usually target localized action, delivery systems impregnated within hydrogels and cardiac patches have gained interest. Injectable hydrogels have been shown to enhance cell survival and attenuate fibrotic responses immediately after myocardial infarction. Ruvinov et al. demonstrated improved cardiac regeneration by facilitating the release of two GFs-IGF-1 (considered cardioprotective) and hepatocyte growth factor (HGF; considered anti-fibrotic) [113]. The group used injectable alginate hydrogel capable of binding to growth factors with affinity-binding AlgS [114]. The injection of the proposed hydrogel into infarcted rat hearts resulted in reduced myocyte apoptosis and fibrosis, and cardiomyocyte proliferation attenuated infarct tissue. Cell-free heart patches have also been extensively studied to effectively induce endogenous cardiac regeneration such as the “paracrine effect.” Cell-based therapies to improve cardiac regeneration involve secretion of cardioprotective factors that signal cells in the infarcted area. In one study, Jeske et al. hypothesized that iPSC- or iPSC-CM-derived EVs (i.e., microvesicles and exosomes) revealed similar effect [115]. The group studied and isolated these EVs in vitro for studying the function of miRNA content. The isolated EVs were encapsulated in a cell-free collagen hydrogel for their tendency to get rapidly consumed by recipient cells, to increase treatment efficacy. Prolonged release of EVs was achieved using a collagen hydrogel for up to 1 week in vivo. In a rat myocardial infarction model, the collagen hydrogel patch reduced scar formation and apoptosis of CMs and enhanced recovery of contractile functions. This system has the advantages of being independent of any cellular component, low risk of immunogenicity and optimal cellular viability and retention. In the following section, spatial regulation is discussed for engineered cardiac patches.

### 5.2. Engineering Vascularized Patch with Spatial Regulation

#### 5.2.1. Engineering Vascularized Patch with Spatial Regulation In Vitro

Multiscale architectures can be precisely engineered using capillary force lithography to recapitulate the topographical and ECM cues. Cellular morphology and directionality can be regulated by nanopatterning of materials through UV-assisted capillary force lithography technique that can impart ECM cues needed. Structures and directions of fibroblast cells have been studied to be predominantly influenced by nano-topography instead of microtopography [116]. Further, micro/nanopatterned transplantable patches composed of PLGA have been fabricated using capillary force lithography/wrinkling combinatory technique. The multiscale PLGA patches displayed augmented tissue adhesion to the underlying native tissue and optimal mechanical strength compared to the merely nanopatterned counterparts [117]. This combinatory strategy can greatly benefit by manipulating the substrate of cell culture to govern cell fate and functionality. Clinically relevant size constructs of hCMPs have been generated. However, patches with relatively larger surface areas (e.g., 8 cm^2^) are comparatively thin (1.25 mm), leading to limited direct perfusion and limited thickness of hCMP to 1–2 mm. Hence, larger and thicker hCMP constructs have been optimized and improved. Engineering thick and viable hCMP generation is limited by inadequate recapitulation of characteristics of native myocardium, such as generation of optimum forces and action potentials. hCMP thickness is limited by the oxygen and nutrients from the vascular network post-transplantation, necessitating the cardiomyocytes to be within 100–200µm distance from the capillaries [118]. hCMPs of optimum thicknesses need the formation of a dense internal vascular network that integrates with native circulation post-transplantation. Vascularization can be enhanced by including a combination of vascular and other cell types (ECs, SMCs, fibroblasts) and also employing nanoparticle-mediated extended release of pro-angiogenic factors, e.g., vascular endothelial growth factor (VEGF), fibroblast growth factor (FGF), and the Wnt activator CHIR99021 for infiltration of the native circulatory loop [119]. In addition, advanced biofabrication methods (e.g., micropatterning and 3D bioprinting) can be used to control the spatial orientation of the vascular network to improve mass transport and perfusion. These examples demonstrate the various techniques that can be used to induce spatial patterned vascularized patches.

#### 5.2.2. Engineering Vascularized Patch with Spatial Regulation In Vivo

In the beginning, tissue printing was employed for generation of constructs human cardiac-derived CPCs and alginate with cardiac lineage commitment and high viability for 7 days in culture [120]. In subsequent experiments, human cardiomyocyte progenitor cells were printed in six perpendicularly printed layers into a hyaluronic acid matrix, and gelatin formed a patch of 4 cm^2^ surface area. Improved measures of cardiac function were observed with increased expression of cardiac and vascular differentiation markers within 4 weeks in a murine MI (myocardial infarction) model [121]. Scaffold-free bioprinted hCMPs have been generated using loaded spheroids onto an array of needles, fused and hCMP cultured as needle holes got filled with surrounding tissue [122]. The construct remained engrafted and displayed vascularization for 7 days after implantation into infarcted rat hearts. Recently, a customized device has been developed for simultaneous loading of layer of spheroids into the needle array, substantially improving print time for larger engineered constructs. An advanced technique called multiphoton-excited (MPE) 3D bioprinting has been employed to improve limitation of low resolution of traditional bioprinting strategies that limit printing structural details to facilitate cellular interactions. This technique controls the architecture of photoactive polymers to reproduce the structural features of the ECM with high fidelity. MPE 3D-printed hCMPs composed of iPSC-derived CMs, ECs, and SMCs in a gelatin scaffold generated calcium transients post fabrication and beat synchronously within 1 day [123]. The printed patches exhibited significant improvements in a murine MI model with enhanced cardiac function (left ventricular ejection fraction and fractional shortening), apoptosis, vascularity, and cell growth. Photoactivated 3D bioprinting has also been studied with a bioink incorporating both ECM proteins and hiPSCs for fabrication of two-chambered structures with both inlet and outlet vessels. Mature cardiac cells were formed from proliferation and differentiation of hiPSCs in situ with human cardiac muscle recapitulating the chambers and large vessels of a human heart [124].

Elaborate research has been conducted for replacement of infarcted cardiac tissue with tissue engineered cardiac patches generated from biocompatible and bioabsorbable components including purified ECM molecules and heterogeneous mixtures of ECM molecules. Jang et al. generated a 3D prevascularized stem cell patch using the spatial arrangement of cardiac progenitor/MSCs with decellularized ECM bio-ink [125]. The cardiac patch reduced fibrosis and cardiac remodeling with enhanced cardiomyogenesis and neovascularization at the injured myocardium post-transplantation. Gao et al. successfully 3D printed an EPC/atorvastatin-loaded PLGA microspheres laden bioink (vascular tissue-derived ECM and alginate) and bio-blood vessel [126]. The engineered tissue exhibited enhanced viability, proliferation, and differentiation of endothelial progenitor cells (EPCs) with improved in vitro endothelialization. The bioblood vessel (BBV)-based technique showed significantly improved EPC function and recovery of ischemic injury in a nude mice hind limb ischemia model. We have previously shown that electrospun aligned microfiber scaffolds could be used for the co-culture of iPSC-derived cardiomyocytes and endothelial cells [78] Upon implantation in vivo, the patches composed of aligned scaffolds induced the formation of microvasculature that were preferentially aligned along the direction of the scaffold microfibers [127] (Figure 3).

In terms of vascularization of patches, apart from infiltration from the native vasculature, thicker hCMPs will likely need engineered vascularity pre-transplantation. Vascularization has been studied to be induced during the fabrication process by encapsulation of a sacrificial gelatin mesh in scaffold material, with subsequent melting of the gelatin mesh to produce a network of interconnected microfluidic channels. The sacrificial scaffolds generated a rudimentary endothelial network when seeded with human microvascular ECs. An alternative strategy using the sustained release of the angiogenic factor thymosin β to promote the outgrowth of vessels from explanted veins and arteries, forming a capillary bed within a hydrogel scaffold mimics the endogenous angiogenic process [128]. Vessel growth can also be induced with micropatterned polyglycerol sebacate scaffolds as they degrade post transplantation with infiltration of host blood cells into the microvessels [129]. Micropatterning has also been employed for organization of ECs into ‘cords’ that induce the capillary for integration with the host tissue post-transplantation. A recent study of 3D printed vessels using thermal inkjet printer for bioprinting human microvascular ECs and fibrin, resulting in generation of micro-sized fibrin channels lined with confluent cells [130]. Vasculature has also been bioprinted with an advanced extrusion platform that generated a sheath of photoactive, cell-laden bioink around an alginate core structure [131]. The alginate was dissolved with a Ca^2+^ chelating agent post UV crosslinking, allowing the cellular population to proliferate and spread forming a perfusable biomimetic vasculature. However, due to limited penetration of UV radiation, in-depth polymerization can be induced via enzymatic reactions, such as the conversion of fibrinogen into fibrin with thrombin as a catalyst [132]. Thick (>1 cm) engineered osteogenic tissues [133] have been generated using the technique of co-printing of vascular and cellular inks in cast ECM material.

## 6. Challenges and Future Prospects

Despite the significant advancements in the field, the generation of thick tissues with functional and mature microvascular networks in vitro faces major challenges for successful clinical tissue/organ translation. Although successful vascularization attempts have made remarkable progress for implantable 3D constructs at clinically relevant scale, building vascular networks that mimic the complexity, microstructure, geometry, biochemical cues, and optimal organ cellular density remains a challenge. Furthermore, the appropriate and timely vascularization of the implanted 3D constructs needs more attention. Direct anastomosis of host microvascular network and preformed microvasculature as reperfusion process for clinically scalable 3D constructs is challenging. Therefore, newer techniques for rapid vascularization are required.

A prominent challenge in angiogenic therapy is the risk of undesired and uncontrolled tissue ingrowth. Significant challenges to fully utilize the potency of angiogenic growth factors therapy include precisely controlling in vivo distribution of growth factor dose and time-duration of bioactivity. Such optimizations are necessary to obviate unstable vessel formation and subsequent regression, hemangioma formation, or neointimal thickening.

Fabrication techniques for sustained release of growth factors from biomimetic scaffolds should render engineered scaffolds with optimal physical properties (e.g., pore size, water content, porosity, the interconnection of pores, etc.) for adequate vascularization. Cells cultured within matrices for generation of vascular structures by themselves result in unwanted architecture and patterns. Advanced techniques like electrospinning, patterning, and 3D printing can be combined to provide a scaffold-guided path for the cellular population [134]. Mimicking the natural hierarchical structure of tissue is imperative for engineered vascularization. Since the effectiveness of bioactive factors vary in vitro and in vivo environments, optimal properties of growth factor delivery from scaffolds should be carefully determined. Along with the need to improve in vitro vascularization strategies is the need to generate immune-evasive cells that can be genetically modified to prevent rejection. Successful implantation and anastomosis of vascularized tissues will further require advanced microsurgical expertise. Although technological advances have now led to the formation of functional and mature vasculature, some challenges remain. Currently, no single vascularization approach can produce a functional, bioactive, stable, and scalable vascular structure, although thin, simple vascular networks have been generated successfully. An optimized approach consisting of a tailored, synergistic combination of several tissue engineering techniques (cells, decellularized tissue, and growth molecules delivery) and inter-disciplinary systems (functionalized biomaterials and fabrication methods) will allow us to engineer improved vascular networks for the development of scalable vascularized 3D tissues.

With respect to the clinical translation of cardiac patches, successful translation will require tissue integration with the surrounding myocardium at three levels: physical and biochemical continuity, electrophysiological cues, and nutrient perfusion. Even though cardiac patches have been successful in improving cell viability and retention, there is a high risk of cells engrafted through such constructs or through intracardiac injection to provoke an immunogenic response, to result in immune rejection of the allograft [135]. In addition, cardiac patch transplantation not accompanied by immunosuppressants can significantly risk transplanted cell survival rate and can cause failing integration. Accessibility for cell migration from areas close to the infarcted zones needs to be allowed by optimal design for formation of blood vessels and nerves to integrate with the host. The presence of a fibrotic scar barrier results in failure of electrical integration with the host. Hence, these limitations must be addressed by cardiac patch designs to be clinically relevant for assimilation with the host myocardium and synchronization over large distances. Despite these challenges, it is envisioned that these limitations will be successfully overcome with time, and the successful implementation of scalable vascularized tissues will become reality in the future.

## Figures and Tables

**Figure 1 bioengineering-09-00555-f001:**
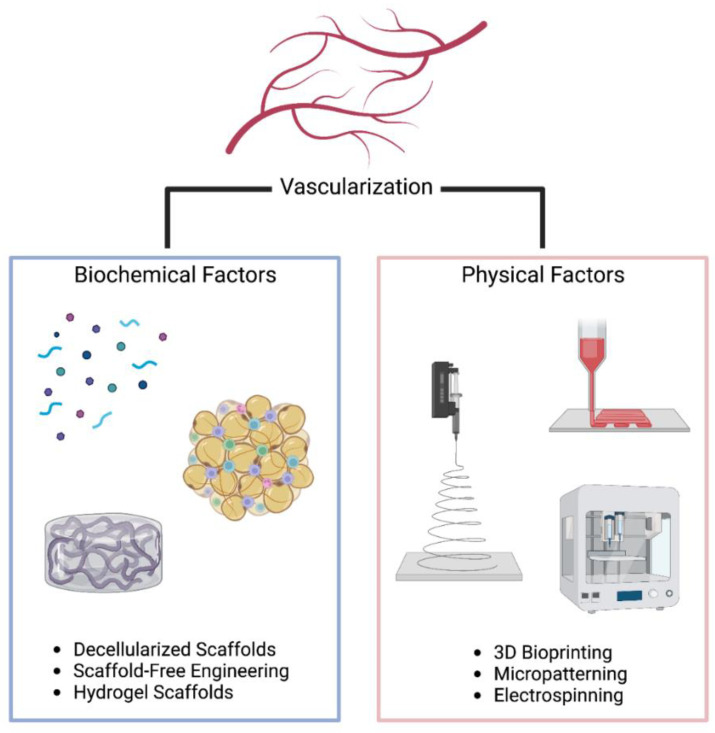
Overview of vascularization strategies using biochemical and biophysical factors.

**Figure 2 bioengineering-09-00555-f002:**
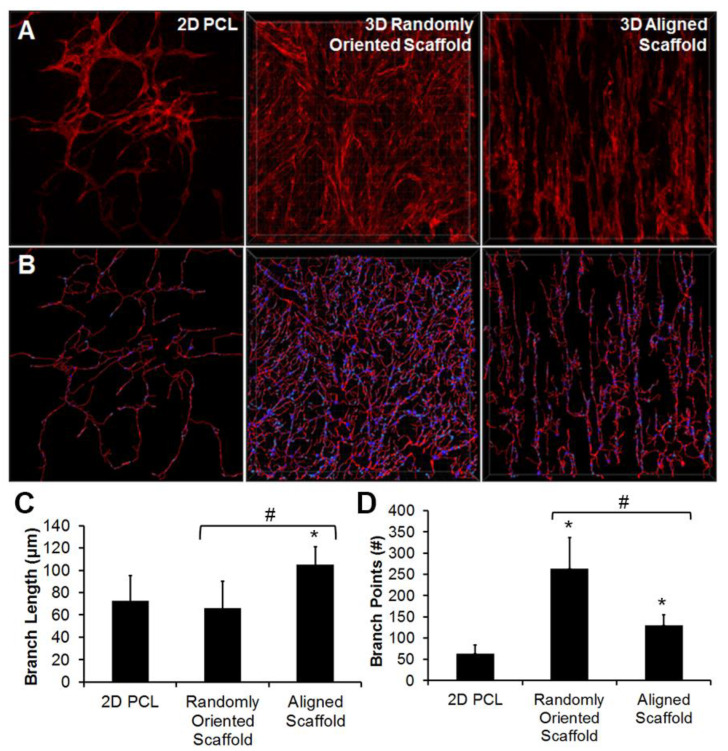
Vascular network-like formation in 3D microfibrous scaffolds. (**A**) 3D stacked confocal images of CD31 staining in 2D PCL film, 3D randomly oriented scaffold, and 3D aligned scaffold. (**B**) Transformation of CD31 expression into skeletonized filaments; (**C**,**D**) Quantification of branch length (**C**) and branch points (**D**). * indicates statistically significant relationship to 2D polycaprolactone (PCL) film, and # indicates statistically significant comparison between 3D groups. *p* < 0.05 (n = 5). Reproduced with permission from [77].

**Figure 3 bioengineering-09-00555-f003:**
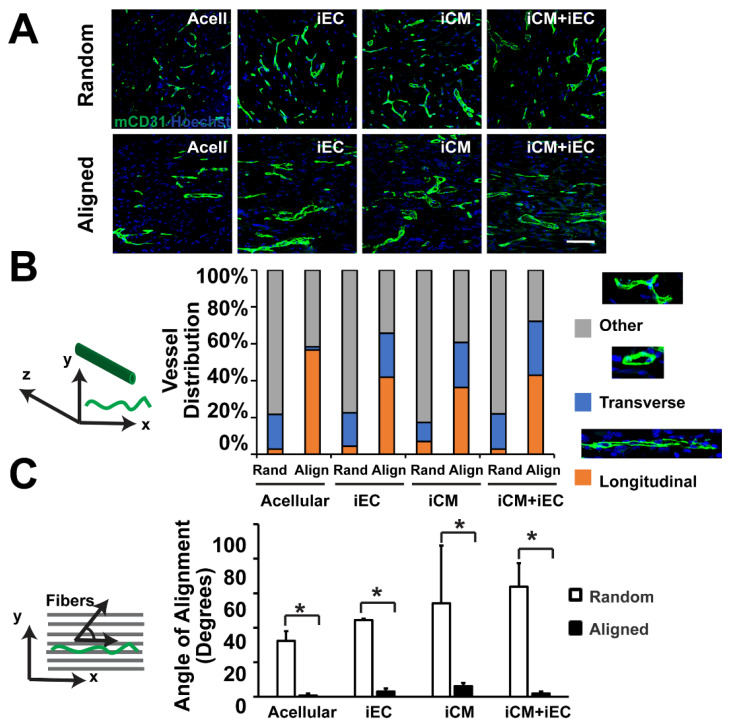
Vascularization of engineered myocardial tissue following subcutaneous implantation into mice. (**A**) Representative confocal microscopy images of CD31 staining (green) within en face sections of engineered myocardial tissues derived from randomly oriented or aligned nanofibrillar scaffolds containing iPSC-derived cardiomyocytes (iCMs), iPSC-derived endothelial cells (iECs), or iCM + iECs at 2 weeks after subcutaneous implantation. Acell denotes acellular scaffold. (**B**) Distribution of vessel orientations within explanted engineered myocardial tissue, relative to the axis of the aligned microfibers as longitudinal, transverse, or other. (**C**) Quantification of the global angle of vessel alignment within subcutaneously explanted engineered myocardial tissues, relative to the axis of the aligned microfibers. The global angle of vessel alignment is calculated as the angle formed by the direction of the longitudinally oriented vessel with respect to the axis of the aligned microfibers. For randomly oriented scaffolds, an arbitrary axis was selected (*n* ≥ 3). Reprinted with permission from [127]. * Denotes statistically significant in comparison (*p* < 0.05).

**Table 1 bioengineering-09-00555-t001:** Bioactive molecules and their effects on vascularization of tissue engineered constructs.

Bioactive Molecules	Angiogenic Effects	Ref
VEGF	Facilitates EC migration and proliferation Regulates EC proliferation, migration, and survival; allows mobilization of BM-derived cells such as HSCs, and recruit SMCs for stabilization of vessel.	[20]
FGF	FGF-2 Enhances EC proliferation. bFGF facilitates the activation, proliferation, and migration of EPC; regulate vasculogenesis and the formation of immature primary vascular networks. FGF-2 Interacts with ECM molecules such as heparin, heparan sulfate proteoglycans (HSPGs); promotes EC response and neovascularization process. FGF-2 facilitates proliferation of ECs, SMCs; endothelial capillary formation	[20]
IGF-1	Facilitates formation of neovasculature from the endothelium of pre-existing vessels andInduces endothelial cell migration for vascularization Induces the activation of the PI3-kinase/Akt signaling pathway and expression of growth factors	[21]
PDGF	Promotes vessel maturation by recruitment of MSCs, pericytes, and SMCs. Facilitates remodeling by inducing collagenases secretion by fibroblasts. Increases VEGF production and promote angiogenesis Regulates the production of ECM molecules for basement membrane and blood vessel stabilization	[22]
TGF-β	Promotes EC migration, proliferation, and differentiation. Increases VEGF secretion by ECs; and PGF and bFGF expression by SMCs. Enhances angiogenesis. Facilitates vessel stabilization and maturation Stimulates ECM deposition	[23]
HGF	Induces VEGF secretion Promotes angiogenesis by ECs expression of VEGF.	[24]
TNF-α	Inhibits proliferation of endothelial cells; promotes angiogenesis	[23]
Angiopoietin	Facilitates TGF-β-induced differentiation of MSCs. Promotes vessel maturation Inhibits VEGF activity and facilitates EC-SMC interactions Enhnaces type IV collagen deposition Promotes EC proliferation Induces VEGF mediated angiogenic sprouting.	[22]
SDF-1	Facilitates vessel stabilization by recruitment of progenitors of SMCs Initiate vascular remodeling; upregulate metalloproteinases and downregulate angiostatin	[25]

Abbreviations: VEGF (Vascular Endothelial growth factor); FGF (Fibroblast growth factor); IGF-1 (insulin-like growth factor); PDGF (Platelet-derived growth factor); TGF-ß (Transforming growth factor-beta); HGF (Hepatocyte-growth factor); SDF-1 (Stromal cell derived growth factor); MSCs (Mesenchymal stem cells); VSMCs (Vascular Smooth muscle cells); TNF-α (Tumor necrosis factor α); EC (Endothelial cells); PGF (Placental growth factor); IGF-1 (Insulin growth factor -1); SMCs (Smooth muscle cells); EC (Endothelial cells); HSPGs (Heparan sulfate proteoglycans); FGF-2 (Fibroblast growth factor -2), bFGF (basic Fibroblast growth factor); HSCs (Hematopoietic stem cells); BM (Bone marrow); PGF (Placental growth factor); ECM (Extracellular matrix).

**Table 3 bioengineering-09-00555-t003:** Multi-material bioprinting strategies for generation of vascularized tissues.

Bioprinting Approach	Targeted Vascularized Tissue	Bioprinter Used	Bioink	Vascularization Impact	Ref
Multi-material bioprinting	Vascularized liver	Double nozzle printing system	ADSC-laden gelatin/alginate/fibrinogen Hepatocytes-laden gelatin/alginate/chitosan	Functional hepatocytes were formed with endothelial like structures in tissue construct.	[60]
	Vascularized bone	3D-bioprinter with two controllable printheads	hMSCs laden gelatin-fibrinogen HUVEC laden gelatin-fibrinogen hydrogel	Osteogenic differentiation factors perfusion through vascular network resulted in osteogenic tissue formation.	[61]
	Vascularized cardiac patch	Multi-head extrusion-based 3D bioprinting	ECs within sacrificial gelatin CMs laden ECM bioink	Heart structure with mechanically stable and robust perfusable vessels	[62]
	Vascularized tissue model	3D bioprinter with more than two controllable printheads	Fibroblast-cell laden GelMA EC injection through microchannels	Fabrication of vascularized tissue constructs.	[63]
Dual 3D bioprinting	SLA-based and extrusion-based bioprinting	Vascularized bone	ECs and hMSCs laden VEGF modified Gel MA- based bioink	Spatial controlled localization of growth factors and perfusion lead to interconnected vascularized bone construct.	[64]
	Extrusion and inkjet bioprinting	Vascularized skin	Adipose-derived dECM and fibrinogen bioink encapsulated human adipocytes Fibroblast cells laden skin dECM and fibrinogen	Formation of vascularized channels between dermis and hypodermis leads to maturation of epidermis with human like structure.	[65]
	Extrusion-based and SLA-based bioprinting platform	Multiphasic hybrid construct vascular conduit model	Cells encapsulated within PEGDA	Diffusion of media into cells resulted in a thick construct	[66]
	Co-axial and extrusion bioprinting platform	Vascular model	Human coronary artery SMCs laden modified Gel MA	Bioprinted vascular construct with biomechanics, perfusion ablility and permeability.	[67]
Co-axial Bioprinting	Coaxial nozzle bioprinting	Vascularized muscle	Endothelial cell-laden vascular dECM	Formation of pre-vascularized muscle with integration into the host tissue and functional recovery.	[68]
	Coaxial nozzle bioprinting	Perfusable renal tissue	Hybrid hydrogel bioink incorporated with kidney dECM and alginate	Renal proximal tube integrated into the host tissues in vivo	[69]
	Coaxial nozzle bioprinting	Vascularized intestinal villi	HUVEC extruded from core region of coaxial nozzle	Human intestine regeneration and organ-on-a-chip system	[70]
	Coaxial bioprinting platform	Vascularized tissue > 1 cm	Cell-laden GelMA Endothelial cell laden gelatin	Generation of tissue models	[71]
Light-based bioprinting	LIFT-Based bioprinting	Vascularized cardiac patch	Deposition of MSCs on a cardiac patch within ECs mesh structure	Pre-vascularized patches with enhanced angiogenesis	[72]
	DLP based-bioprinting	Vascularized thick tissue	Photopolymerizable glycidyl methacrylate- hyaluronic acid and GelMA	Fabrication of vascularized tissue constructs with high resolution.	[73]

Abbreviations: EC (Endothelial cells); dECM (decellularized extracellular matrix); MSCs (Mesenchymal stem cells); Gel MA (Gelatin methacryloyl); HUVEC (Human vascular endothelial cells); PEGDA (Polyethylene glycol diacrylate); DLP (Digital Light Processing); SMCs (Smooth muscle cells); VEGF (Vascular endothelial growth factor); LIFT (Laser-induced forward transfer); CMs (Cardiomyocytes); ADSC (Adipose derived stem cells); SLA (Stereolithography).

**Table 4 bioengineering-09-00555-t004:** Fabrication techniques/structures to promote vascularization in tissue engineered constructs.

Technique/Structures	Application	Limitations	Ref
3D Bioprinting	Developed tissue constructs mimic the spatial, mechanochemical, and temporal characteristics of native tissues Microchannels of width > 100 µm can be obtained Heterogenous tissues constructs can be created (>1 cm in thickness and 10 cm3 volume). Multicellular spheroids (>400 µm diameter) are bioprinted and double layered small diameter conduits of diameter 2.5 mm. High accuracy and reproducibility High precision in 3D structure Modularity of bio-inks	Print resolution is limited Print size limited to diffusion	[94]
Micropatterning	Promotes cell alignment and cell density High reproducibility Can be integrated with other techniques.	Limited complexity of organized tissue. Constructs unable to be implanted Size scale is limited	[95]
Hydrogel	Biocompatible Can match tissue stiffness	Limited cell directionality Fragile construct	[96]
Electrospinning	High reproducibility Relatively low cost Cellular alignment maintained	Low biocompatibility Limited tissue complexity.	[97]
Decellularized Scaffolds	Recapitulate 3D organ specific architecture Native vascular network is largely preserved Low cytotoxicity	Limited efficiency Limited tissue/organ donor availability Antigenicity from xenogenic tissues	[98]
Tissue Engineered Heart	Constructs have native myocardial structure Cardiomyocyte contractility is maintained	Low apparatus modularity Restricted applications	[99]
Scaffold-free Engineering	High reproducibility and efficiency Physiological Cell–cell interaction Controlled growth factor release	Limited accessibility Restricted applications. Lack of precision in network architecture	[100]

Abbreviations: 3D (Three dimensional).

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
