# Peer review of "Engineering Spatiotemporal Control in Vascularized Tissues"

_bioengineering, 2022, doi:10.3390/bioengineering9100555_

Round 1

Reviewer 1 Report

This review manuscript by Khanna et al. reports the recent developments towards vascularized tissue engineering, with a special focus on the fabrication methods, role of angiogenic molecules, spatiotemporal control and vascular cardiac patches (which is the main expertise of the group). In brief, the first two introductory sections   about temporal biology of angiogenesis and growth factors regulation are very interesting, quite concise and quite original (in comparison with the plethora of reviews in the field). This review is also well documented with more than 130 adequate citations. There is a wealth on information in this manuscript. However, in my opinion, for a review paper to have a real impact, it has to be to bring a comprehensive and synthetic view instead of being only a report of works one after the others. This is the main criticism. The authors mostly accumulate the description of works one after the other instead of gathering them by type of approach or system. As a consequence, the manuscript is a bit wordy and not so exciting to read. Instead of rather indigestible tables (often with absent separations between two consecutive works, so that one does not know which application applies to which work (eg. table 2)). I think that this type of reviewing work that aims at identifying the future challenges in the field would highly benefit of figures/synthetic drawing (more than Figure 1, which is uninformative or very common). There is also huge number of typos (with extra or m issing comas, periods, letters – see eg: line 151 (nregulate), lines 153, 297 (D Bioprinting), 364 (eterogeneous), 401 (viscoud), 616, 637, 669, 736, 762 (vito)). I then expected to find a nice synthesis in the conclusion about the future challenges in vascularized tissues. But here again, past works are cited, and the authors seemed to have omitted most references (especially in paragraphs from line 769 to 801).

Altogether, nothing wrong in this review and a nice bibliographic work. However, due to its lack of insightful synthesis,  I am not convinced it will be widely cited. Still, I have nothing against its publication.

Author Response

We thank the reviewer for the positive feedback on this review manuscript. We have now addressed the major concerns as described below:

  1. Comprehensive View & Synthesis in the Conclusion: We have now restructured the Conclusion section to focus on future challenges, without the inclusion of excessive citations.
  2. Table Organization: As suggested by the reviewer, we now include separations between major subheadings for improved clarity.
  3. Figures: We now have added Figures 2 and 3 to enhance the visual quality of the manuscript.
  4. Typos: we have now proof-read the manuscript to eliminate typos.

Reviewer 2 Report

The review focuses on cardiac and other vascularized tissues. The importance of vascularization in these types of tissues is very well described as well as the different types of scaffold. The bibliographic references are relevant and almost complete on the subject.

The only downside is that the part on iPS and the role of cells obtained by differentiating them in tissues has not been more addressed. Indeed, the current trend, in tissues as well for toxicology models, disease or tissue repair, is to generate scaffold with a population of isogenic cells from iPS. A recent article by Mummery's group published in 2020 addresses this topic. It would have been wise to add it to the bibliography and thus complete this part of the review a little thin.

There is one typograhic errors in the text:

-          Two “of “ line 557.

Author Response

We thank the reviewer for the positive feedback on this review manuscript. We have now addressed the major concerns as described below:

iPSC and role of cells obtained by differentiating them in tissues: The advantages and disadvantages of iPSCs for tissue engineering is now described.

Reference: The reference for work by Mummery et.al. has been added.

Reviewer 3 Report

The authors provided a comprehensive review on the recent developments and methods in biofabrication (electrospinning, micropatterning, and 3D bioprinting) techniques for engineered tissues. Furthermore, the review also described the temporal control advancements of growth factors and drugs that are involved in key biological processes, including stem cell differentiation and functional tissue regeneration. Lastly, some key challenges and limitations of the current biofabrication techniques for tissue engineering are also discussed.

Overall, the review is well-written and the scope of the review is well-covered. The authors also described key challenges faced in this area of research and suggested some possible area of work that could advance this field of research. Minor spelling errors (eg: line 307) and extra blank spaces (line 207, 209) are spotted throughout the manuscript, please kindly check.

Author Response

We thank the reviewer for the positive feedback on this review manuscript. We have now addressed the major concerns as described below:

Spelling errors: The spelling errors and blank spaces have been corrected.